# STATE-DENOISED RECURRENT NEURAL NETWORKS

## ABSTRACT

Recurrent neural networks (RNNs) are difficult to train on sequence processing tasks, not only because input noise may be amplified through feedback, but also because any inaccuracy in the weights has similar consequences as input noise. We describe a method for denoising the hidden state during training to achieve more robust representations thereby improving generalization performance. Attractor dynamics are incorporated into the hidden state to 'clean up' representations at each step of a sequence. The attractor dynamics are trained through an auxillary denoising loss to recover previously experienced hidden states from noisy versions of those states. This *state-denoised recurrent neural network* (*SDRNN*) performs multiple steps of internal processing for each external sequence step. On a range of tasks, we show that the SDRNN outperforms a generic RNN as well as a variant of the SDRNN with attractor dynamics on the hidden state but without the auxillary loss. We argue that attractor dynamics—and corresponding connectivity constraints—are an essential component of the deep learning arsenal and should be invoked not only for recurrent networks but also for improving deep feedforward nets and intertask transfer.

## 1 INTRODUCTION

Noise robustness is a fundamental challenge for every information processing system. A traditional approach to handling noise is to design preprocessing stages that estimate and filter noise from an input signal (Boll, 1979). More recently in machine learning, loss functions have been explored to achieve invariance to task-irrelevant perturbations in the input (Simard et al., 1992; Zheng et al., 2016). Although such methods are suitable for handling external noise, we argue in this article that *internal* noise can be an even greater challenge. To explain what we mean by internal noise, consider a deep-net architecture in which representations are transformed from one hidden layer to the next. To the degree that the network weights are not precisely tuned to extract only the critical features of the domain, irrelevant features may be selected and may potentially interfere with subsequent processing. Recurrent architectures are particularly fragile because internal-state dynamics can amplify noise over the course of sequence processing (Anonymous, 1994). In this article, we propose a suppression method that improves the generalization performance of deep nets. We focus on recurrent nets, where the potential benefits are most significant.

Our approach draws inspiration from the human brain, a notably robust and flexible information processing system. Although the brain operates in an extremely high-dimensional continuous state space, the conscious mind categorizes and interprets the world via language. We argue that categorization and interpretation processes serve to suppress noise and increase the reliability of behavior. Categorization treats instances that vary in incidental ways as identical: a chair is something you can sit on regardless of whether it is made of wood or metal. Language plays a similar role in cognition. For example, Witthoft et al. (2003) demonstrated that color terms of one's native language influence a perceptual-judgment task—selecting a color patch that best matches a reference patch. The linguistic label assigned to a patch is the basis of matching, not the low-level perceptual data.

Suppressing variability facilitates *communication* in information-processing systems. Whether we are considering groups of humans, regions within the brain, components of an articulated neural net architecture, or layers of a feedforward network, information must be communicated from a *sender* to a *receiver*. To ensure that the receiver interprets a message as intended, the sender should limit its messages to a canonical form that the receiver has interpreted successfully in the past. Language

Figure 1: (a) energy landscape, (b) attractor net, (c) attractor net unfolded in time.

serves this role in inter-human communication. We explore noise suppression methods to serve this role in intra-network communication in deep-learning models.

In the next section, we describe a recurrent neural network architecture for cleaning up noisy representations—an *attractor net*. We then propose integrating attractor networks into deep networks, specifically sequence-processing networks, for denoising internal states. We present a series of experiments on small-scale problems to illustrate the methodology and analyze its benefits, followed by experiments on more naturalistic problems which also show reliable and sometimes substantial benefits of denoising, particularly in data-limited situations. We postpone a discussion of related work until we present the technical content of our work, which should facilitate a comparison.

## 2 NOISE SUPPRESSION VIA ATTRACTOR DYNAMICS

The attractor nets we explore are discrete-time nonlinear dynamical systems. Given a static $n$-dimensional input $c$, the network state at iteration $k$, $a_k$, is updated according to:

$$a_k = f\left(W a_{k-1} + c\right),$$ (1)

where $f$ is a nonlinearity and $W$ is a weight matrix. Under certain conditions on $f$ and $W$, the state is guaranteed to converge to a *limit cycle* or *fixed point*. A limit cycle of length $\lambda$ occurs if $\lim_{k\to\infty} a_k = a_{k+\lambda}$. A fixed point is the special case of $\lambda = 1$.

Attractor nets have a long history starting with the work of Hopfield (1982) that was partly responsible for the 1980s wave of excitement in neural networks. Hopfield defined a mapping from network state, $a$, to scalar *energy* values via an energy (or Lyapunov) function, and showed that the dynamics of the net perform local energy minimization. The shape of the energy landscape is determined by weights $W$ and input $c$ (Figure 1a). Hopfield's original work is based on binary-valued neurons with asynchronous update; since then similar results have been obtained for certain nets with continuous-valued nonlinear neurons acting in continuous (Hopfield, 1984) or discrete time (Koiran, 1994). We adopt Koiran's 1994 framework, which dovetails with the standard deep learning assumption of synchronous updates on continuous-valued neurons. Koiran shows that with symmetric weights ($w_{ji} = w_{ij}$), nonnegative self-connections ($w_{ii} \geq 0$), and a bounded nonlinearity $f$ that is continuous and strictly increasing except at the extrema (e.g., tanh, logistic, or their piece-wise linear approximations), the network asymptotically converges over iterations to a fixed point or limit cycle of length 2. As a shorthand, we refer to convergence in this manner as *stability*.

Attractor nets have been used for multiple purposes, including content-addressable memory, information retrieval, and constraint satisfaction (Mozer, 2009; Siegelmann, 2008). In each application, the network is given an input containing partial or corrupted information, which we will refer to as the *cue*, denoted $c$; and the cue is mapped to a canonical or *well-formed* output in $a_\infty$. For example, to implement a content-addressable memory, a set of vectors, $\{\xi^{(1)}, \xi^{(2)}, \ldots\}$, must first be stored. The energy landscape is sculpted to make each $\xi^{(i)}$ an attractor via a supervised training procedure in which the target output is the vector $\xi^{(i)}$ for some $i$ and the input $c$ is a noise-corrupted version of the target. Following training, noise-corrupted inputs should be 'cleaned up' to reconstruct the state.

In the model described by Equation 1, the attractor dynamics operate in the same representational space as the input (cue) and output. Historically, this is the common architecture. By projecting the input to a higher dimensional latent space, we can design attractor nets with greater representational capacity. Figure 1b shows our architecture, with an $m$-dimensional input, $x \in [-1, +1]^m$, an $m$-dimensional output, $y \in [-1, +1]^m$, and an $n$-dimensional attractor state, $a$, where typically $n > m$ for representational flexibility. The input $x$ is projected to the attractor space via an affine transform:

$$c = W^{\text{IN}} x + v^{\text{IN}}.$$ (2)

The attractor dynamics operate as in Equation 1, with initialization $\boldsymbol{a}_0 = \boldsymbol{0}$ and $f \equiv \tanh$. Finally, the asymptotic attractor state is mapped to the output:

$$\boldsymbol{y} = f^{\mathrm{OUT}}\left(\boldsymbol{W}^{\mathrm{OUT}}\boldsymbol{a}_\infty + \boldsymbol{v}^{\mathrm{OUT}}\right), \tag{3}$$

where $f^{\mathrm{OUT}}$ is an output activation function and the $\boldsymbol{W}^*$ and $\boldsymbol{v}^*$ are free parameters of the model.

To conceptualize a manner in which this network might operate, $\boldsymbol{W}^{\mathrm{IN}}$ might copy the $m$ input features forward and the attractor net might use $m$ degrees of freedom in its state representation to maintain these *visible* features. The other $n - m$ degrees of freedom could be used as *latent* features that impose higher-order constraints among the visible features (in much the same manner as the hidden units in a restricted Boltzmann machine). When the attractor state is mapped to the output, $\boldsymbol{W}^{\mathrm{OUT}}$ would then transfer only the visible features.

As Equations 1 and 2 indicate, the input $\boldsymbol{x}$ biases the attractor state $\boldsymbol{a}_k$ at every iteration $k$, rather than—as in the standard recurrent net architecture—being treated as an initial value, e.g., $\boldsymbol{a}_0 = \boldsymbol{x}$. Effectively, there are short-circuit connections between input and output to avoid vanishing gradients that arise in deep networks (Figure 1c). As a result of the connectivity, it is trivial for the network to copy $\boldsymbol{x}$ to $\boldsymbol{y}$—and thus to propagate gradients back from $\boldsymbol{y}$ to $\boldsymbol{x}$. For example, the network will implement the mapping $\boldsymbol{y} = \boldsymbol{x}$ if: $m = n$, $\boldsymbol{W}^{\mathrm{IN}} = \boldsymbol{W}^{\mathrm{OUT}} = \boldsymbol{I}$, $\boldsymbol{v}^{\mathrm{IN}} = \boldsymbol{v}^{\mathrm{OUT}} = \boldsymbol{0}$, $\boldsymbol{W} = \boldsymbol{0}$, and $f^{\mathrm{OUT}}(\boldsymbol{z}) = \boldsymbol{z}$ or $f^{\mathrm{OUT}}(\boldsymbol{z}) = \max(-1, \min(+1, \boldsymbol{z}))$.

In our simulations, we use an alternative formulation of the architecture that also enables the copying of $\boldsymbol{x}$ to $\boldsymbol{y}$ by treating the input as unbounded and imposing a bounding nonlinearity on the output. This variant consists of: the input $\boldsymbol{x}$ being replaced with $\hat{\boldsymbol{x}} \equiv \tanh^{-1}(\boldsymbol{x})$ in Equation 2, $f^{\mathrm{OUT}} \equiv \tanh$ in Equation 3, and the nonlinearity in the attractor dynamics being shifted back one iteration, i.e., Equation 1 becomes $\boldsymbol{a}_k = \boldsymbol{W}f(\boldsymbol{a}_{k-1}) + \boldsymbol{c}$. This formulation is elegant if $\boldsymbol{x}$ is the activation pattern from a layer of tanh neurons, in which case the tanh and $\tanh^{-1}$ nonlinearities cancel. Otherwise, to ensure numerical stability, one can define the input $\hat{\boldsymbol{x}} \equiv \tanh^{-1}[(1 - \epsilon)\boldsymbol{x}]$ for some small $\epsilon$.

## 2.1 TRAINING A DENOISING ATTRACTOR NETWORK

We demonstrate the supervised training of a set of attractor states, $\{\boldsymbol{\xi}^{(1)}, \boldsymbol{\xi}^{(2)}, \ldots \boldsymbol{\xi}^{(A)}\}$, with $\boldsymbol{\xi}^{(i)} \sim \mathrm{Uniform}(-1, +1)^m$. The input to the network is a noisy version of some state $i$, $\hat{\boldsymbol{x}}^{(i)} = \tanh^{-1}(\boldsymbol{\xi}^{(i)}) + \boldsymbol{\eta}$, with $\boldsymbol{\eta} \sim \mathcal{N}(\boldsymbol{0}, \sigma^2 I)$, and the corresponding target output is simply $\boldsymbol{\xi}^{(i)}$. With $\kappa$ noisy instances of each attractor for training, we define a normalized denoising loss

$$\mathcal{L}_{\mathrm{denoise}} = \frac{1}{\kappa A} \sum_{i=1}^{\kappa A} \frac{||\boldsymbol{y}^{(i)} - \boldsymbol{\xi}^{(i)}||^2}{||\tanh(\hat{\boldsymbol{x}}^{(i)}) - \boldsymbol{\xi}^{(i)}||^2}, \tag{4}$$

to be minimized by stochastic gradient descent. The aim is to sculpt attractor basins whose diameters are related to $\sigma^2$. The normalization in Equation 4 serves to scale the loss such that $\mathcal{L}_{\mathrm{denoise}} \geq 1$ indicates failure of denoising and $\mathcal{L}_{\mathrm{denoise}} = 0$ indicates complete denoising. The $[0, 1]$ range of this loss helps with calibration when combined with other losses.

We trained attractor networks varying the number of target attractor states, $A \in [25, 250]$, the noise corruption, $\sigma \in \{0.125, 0.250, 0.500\}$, and the number of units in the attractor net, $n \in [25, 250]$. In all simulations, the input and output dimensionality is fixed at $m = 50$, $\kappa = 50$ training inputs and $\kappa = 50$ testing inputs were generated for each of the $A$ attractors. The network is run to convergence. Due to the possibility of a limit cycle of 2 steps, we used the convergence criterion $||\boldsymbol{y}_{k+2} - \boldsymbol{y}_k||_\infty < \delta$, where $\boldsymbol{y}_k$ is the output at iteration $k$ of the attractor dynamics. This criterion ensures that no element of the state is changing by more than some small $\delta$. For $\delta = .01$, we found convergence typically in under 5 steps, nearly always under 10.

Figure 2 shows the percentage of noise variance removed by the attractor dynamics on the test set, defined as $100(1 - \mathcal{L}_{\mathrm{denoise}})$. The three panels correspond to different levels of noise in the training set; in all cases, the noise in the test set was fixed at $\sigma = 0.250$. The 4 curves each correspond to a different size of the attractor net, $n$. Larger $n$ should have greater capacity for storing attractors, but also should afford more opportunity to overfit the training set. For all noise levels, the smallest ($n = 50$) and largest ($n = 200$) nets have lower storage capacity than the intermediate ($n = 100, 150$) nets, as reflected by a more precipitous drop in noise suppression as $A$, the number of attractors to be stored, increases. One take-away from this result is that roughly we should choose $n \approx 2m$, that

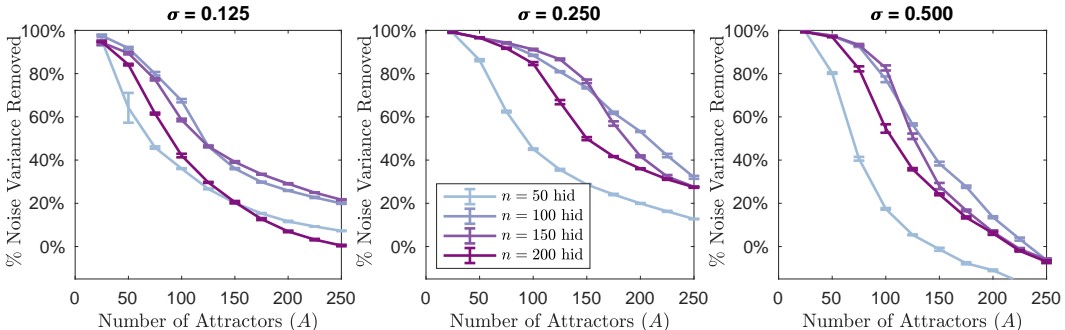

Figure 2: Percentage noise variance suppression by an attractor net trained on 3 noise levels ($\sigma$). In each graph, the number of hidden units in the attractor net ($n$) and number of attractors to be learned ($A$) is varied. In all cases, inputs are 50-dimensional and evaluation is based on a fixed $\sigma = 0.250$. Ten replications are run of each condition; error bars indicate $\pm 1$ SEM.

is, the hidden attractor space should be about twice as large as the input/output space. This result does not appear to depend on the number of attractors stored, but it may well depend on the volume of training data. Another take-away from this simulation is that the noise level in training should match that in testing: training with less ($\sigma = 0.125$) and more ($\sigma = 0.500$) noise than in testing ($\sigma = 0.250$) resulted in poorer noise suppression. Based on this simulation, the optimal parameters, $\sigma = 0.250$ and $n = 100$, should yield a capacity for the network that is roughly $2n$, as indicated by the performance drop that accelerates for $A > 100$.

## 3 STATE-DENOISED RECURRENT NEURAL NETWORKS

After demonstrating that attractor dynamics can be used to denoise a vector representation, the next question we investigate is whether denoising can be applied to internal states of a deep network to improve its generalization performance. As we explained earlier, our focus is on denoising the hidden state of recurrent neural networks (RNNs) for sequence processing. RNNs are particularly susceptible to noise because feedback loops can amplify noise over the course of a sequence.

Figure 3 shows an unrolled RNN architecture with an integrated attractor net. The iterations of the attractor net are unrolled vertically in the Figure, and the steps of the sequence are unrolled horizontally, with each column indicating a single step of the sequence with a corresponding input and output. The goal is for the hidden state to be denoised by the attractor net, yielding a cleaned hidden state, which then propagates to the next step. In order for the architecture to operate as we describe, the parameters of the attractor net (blue and green connections in Figure 3) must be trained to minimize $\mathcal{L}_{\text{denoise}}$ and the parameters of the RNN (purple, orange, and red connections in Figure 3) must be trained to minimize a supervised *task loss*, $\mathcal{L}_{\text{task}}$, defined over the outputs. We have found it generally beneficial to also train the parameters of the attractor net on $\mathcal{L}_{\text{task}}$, and we describe this version of the model.

We refer to the architecture in Figure 3 as a *state denoised recurrent neural net* (or *SDRNN*) when it is trained on the two distinct objectives, $\mathcal{L}_{\text{denoise}}$ and $\mathcal{L}_{\text{task}}$, and as a *recurrent neural net with attractors* (or *RNN+A*) when all parameters in the model are trained solely on $\mathcal{L}_{\text{task}}$. For each minibatch of training data, the SDRNN training procedure consists of:

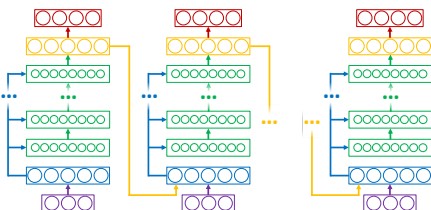

Figure 3: SDRNN architecture for sequence processing, unrolled in time such that each column denotes a single time step with a corresponding input and output. The hidden state is denoised by the attractor net, yielding a cleaned state which is combined with the next input to determine the next hidden state.

1. Take one stochastic gradient step on $\mathcal{L}_{\text{task}}$ for all RNN and attractor weights.

2. Recompute hidden state for each training sequence $s$ and each step of the sequence $t$, $H = \{\boldsymbol{h}^{(s,t)}\}$.

3. With just the attractor net, take one stochastic gradient step on $\mathcal{L}_{\text{denoise}}$ for the attractor weights using the procedure in Section 2.1 with $\boldsymbol{\xi} \in H$.

4. Optionally, step 3 might be repeated until $\mathcal{L}_{\text{denoise}} < 1$.

# 4 SIMULATIONS

## 4.1 PARITY TASK

We studied a streamed *parity* task in which 10 binary inputs are presented in sequence and the target output following the last sequence element is $1$ if an odd number of 1s is present in the input or $0$ otherwise. The architecture has $m = 10$ hidden units, $n = 20$ attractor units, and a single input and a single output. We experimented with both tanh and GRU hidden units. We trained the attractor net with $\sigma = .5$ and ran it for exactly 15 iterations (more than sufficient to converge). Models were trained on 256 randomly selected binary sequences. Two distinct test sets were used to evaluate models: one consisted of the held-out 768 binary sequences, and a second test set consisted of three copies of each of the 256 training sequences with additive uniform $[-0.1, +0.1]$ input noise. We performed one hundred replications of a basic RNN, the SDRNN, and the RNN+A (the architecture in Figure 3 trained solely on $\mathcal{L}_{\text{task}}$). Other details of this and subsequent simulations are presented in the Supplementary Materials.

Figure 4a shows relative performance on the held-out sequences by the RNN, RNN+A, and SDRNN with a tanh hidden layer. Figure 4b shows the same pattern of results for the noisy test sequences. The SDRNN significantly outperforms both the RNN and the RNN+A: it generalizes better to novel sequences and is better at ignoring additive noise in test cases, although such noise was absent from training. Figures 4c,d show similar results for models with a GRU hidden layer. Absolute performance improves for all three recurrent net variants with GRUs versus tanh hidden units, but the relative pattern of performance is unchanged. Note that the improvement due to denoising the hidden state (i.e., SDRNN vs. RNN for both tanh and GRU architectures) is much larger than the improvement due to switching hidden unit type (i.e., RNN with GRUs vs. tanh hidden), and that the use of GRUs—and the equivalent LSTM—is viewed as a critical innovation in deep learning.

In principle, parity should be performed more robustly if a system has a highly restricted state space. Ideally, the state space would itself be binary, indicating whether the number of inputs thus far is even or odd. Such a restricted representation should force better generalization. Indeed, quantizing

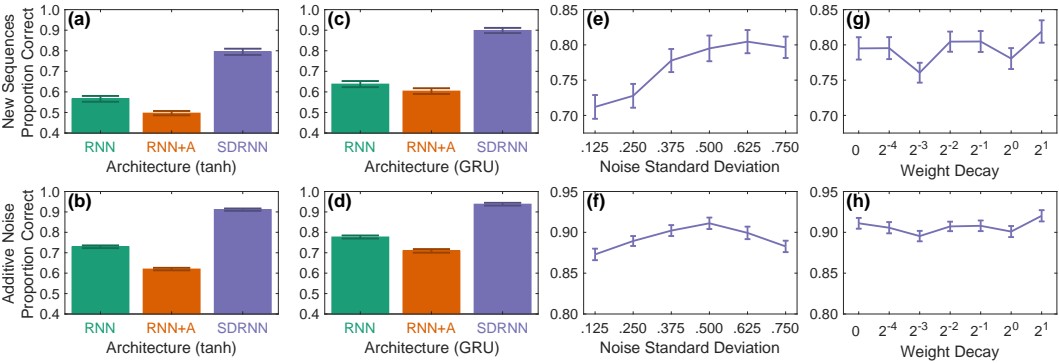

Figure 4: Parity simulations. Top row shows generalization performance on novel binary sequences; bottom row shows performance on trained sequences with additive noise. Unless otherwise noted, the simulations of the SDRNN use $\sigma = 0.5$, 15 attractor iterations, and $L_2$ regularization (a.k.a. weight decay) 0.0. Error bars indicate $\pm 1$ SEM, based on a correction for confidence intervals with matched comparisons (Masson and Loftus, 2003).

the hidden activation space for all sequence steps of the test set, we obtain a lower entropy for the tanh SDRNN (3.70, standard error .06) than for the tanh RNN (4.03, standard error .05). However, what is surprising about this simulation is that gradient-based procedures could learn such a restricted representation, especially when two orthogonal losses compete with each other during training. The competing goals are clearly beneficial, as the RNN+A and SDRNN share the same architecture and differ only in the addition of the denoising loss.

Figure 4e,f show the effect of varying $\sigma$ in training attractors. If $\sigma$ is too small, the attractor net will do little to alter the state, and the model will behave as an ordinary RNN. If $\sigma$ is too large, many states will be collapsed together and performance will suffer. An intermediate $\sigma$ thus must be best, although what is 'just right' should turn out to be domain dependent. We explored one additional manipulation that had no systematic effect on performance. We hypothesized that because the attractor-net targets change over the course of learning, it might facilitate training to introduce weight decay in order to forget the attractors learned early in training. We introduced an $L_2$ regularizer using the ridge loss, $\mathcal{L}_{\text{ridge}} = \lambda ||\boldsymbol{W}||_2^2$, where $\boldsymbol{W}$ is the symmetric attractor weight matrix in Equation 1 and $\lambda$ is a weight decay strength. As Figures 4g,h indicate, weight decay has no systematic effect, and thus, all subsequent experiments use a ridge loss $\lambda = 0$.

The noise being suppressed during training is neither input noise nor label noise; it is noise in the internal state due to weights that have not yet been adapted to the task. Nonetheless, denoising internal state during training appears to help the model overcome input noise and generalize better.

## 4.2 MAJORITY TASK

We next studied a *majority* task in which the input is a binary sequence and the target output is 1 if a majority of inputs are 1, or 0 otherwise. We trained networks on 100 distinct randomly drawn fixed-length sequences, for length $l \in 11, 17, 23, 29, 35$. We performed 100 replications for each $l$ and each model. We ensured that runs of the various models were matched using the same weight initialization and the same training and test sets. All models had $m = 10$ tanh hidden units, $n = 20$ attractor units, $\sigma = .25$.

We chose the majority task because, in contrast to the parity task, we were uncertain if a restricted state representation would facilitate task performance. For the majority task of a given length $l$, the network needs to distinguish roughly $2l$ states. Collapsing them together is potentially dangerous: if the net does not keep exact count of the input sequence imbalance between 0's and 1's, it may fail.

As in the parity task, we tested both on novel binary sequences and training sequences with additive uniform noise. Figures 5a,b show that neither the RNN nor RNN+A beats the SDRNN for any sequence length on either test set. The SDRNN seems superior to the RNN for short novel sequences and long noisy sequences. For short noisy sequences, both architectures reach a ceiling. The only disappointment in this simulation is the lack of a difference for novel long sequences.

## 4.3 REBER GRAMMAR

The Reber grammar (Reber, 1967), shown in Figure 5c, has long been a test case for artificial grammar learning (e.g., Hochreiter and Schmidhuber, 1997). The task involves discriminating between strings

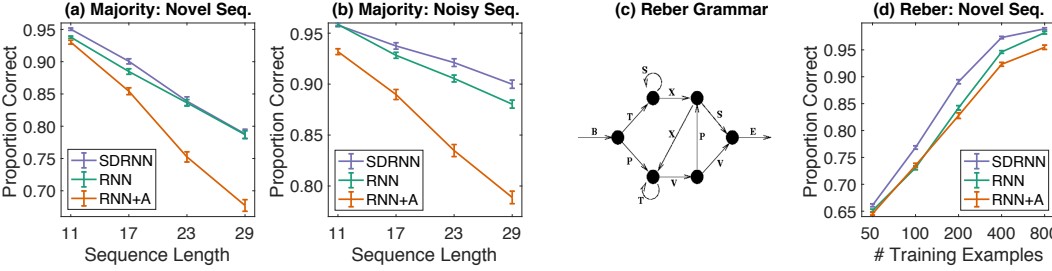

Figure 5: Simulation results on majority task with (a) novel and (b) noisy sequences. (c) Reber grammar. (d) Simulation results on Reber grammar. Error bars indicate $\pm 1$ SEM, based on a correction for confidence intervals with matched comparisons (Masson and Loftus, 2003).

that can and cannot be generated by the finite-state grammar. We generated positive strings by sampling from the grammar with uniform transition probabilities. Negative strings were generated from positive strings by substituting a single symbol for another symbol such that the resulting string is out-of-grammar. Examples of positive and negative strings are BTSSXXTTVPSE and BPTVPXTSPSE, respectively. Our networks used a one-hot encoding of the seven input symbols, $m = 20$ tanh hidden units, $n = 40$ attractor units, and $\sigma = 0.25$. The number of training examples was varied from 50 to 800, always with 2000 test examples. Both the training and test sets were balanced to have an equal number of positive to negative strings. One hundred replications of each simulation was run.

Figure 5d presents mean test set accuracy on the Reber grammar as a function of the number of examples used for training. As with previous data sets, the SDRNN strictly outperforms the RNN, which in turn outperforms the RNN+A.

### 4.4 SYMMETRY TASK

The symmetry task involves detecting symmetry in fixed-length symbol strings such as ACAFBBFACA. This task is effectively a memory task for an RNN because the first half of the sequence must be retained to compare against the second half. We generated strings of length $2s + f$, where $s$ is the number of symbols in the left and right sides and $f$ is the number of intermediate fillers. For $i \in \{1, ..., s\}$, we generated symbols $S_i \in \{A, B, ..., H\}$. We then formed a string $X$ whose elements are determined by $S$: $X_i = S_i$ for $i \in \{1, ..., s\}$, $X_i = \emptyset$ for $i \in \{s + 1, ..., s + f\}$, and $X_i = S_{2s+f+1-i}$ for $i \in \{s + f + 1, ..., 2s + f\}$. The filler $\emptyset$ was simply a unique symbol. Negative cases were generated from a randomly drawn positive case by either exchanging two adjacent distinct non-null symbols, e.g., ACAFBBAFCA, or substituting a single symbol with another, e.g., AHAFBBFACA. Our training and test sets had an equal number of positive and negative examples, and the negative examples were divided equally between the sequences with exchanges and substitutions.

We trained on 5000 examples and tested on an additional 2000, with the half sequence having length $s = 5$ and with an $f = 1$ or $f = 10$ slot filler. The longer filler makes temporal credit assignment more challenging. As shown in Figures 6a,b, the SDRNN obtains as much as a 70% reduction in test error over either the RNN or RNN+A.

### 4.5 PART-OF-SPEECH TAGGING

In natural language processing, part-of-speech (POS) tagging assigns a label to each word in a sentence. Because words change their role depending on context (e.g. "run" can be a noun or a verb), the task requires recognizing how the context influences the label. We used a data set from the NLTK library (Bird et al., 2009). Each sentence is a sequence, presented one word at a time. Words are represented using a 100-dimensional pretrained GloVe embedding (Pennington et al., 2014), and the 147 most popular POS tags are used as outputs (with an additional catch-all category) in a one-hot representation. Models are trained with a varying number of examples and tested on a fixed 17,202 sentences. Early stopping is performed with a validation set which is 20% of the training set. We compare SDRNN and RNN on four replications of each simulation; in each replication the same subsets of training/validation data are used.

The basic RNN architecture is a single-layer bidirectional GRU network (Chung et al., 2015) with a softmax output layer. The SDRNN incorporates one attractor network for the forward GRU and another for the backward GRU. We use $m = 50$ GRUs in each direction and $n = 100$ attractor units in each network. Additional details of the training procedure are presented in the Supplementary

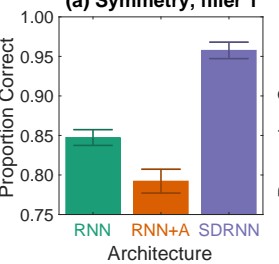

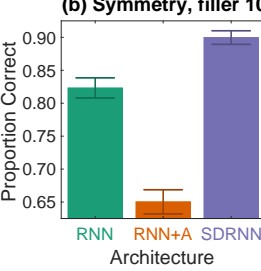

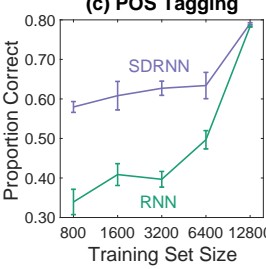

Figure 6: (a) Symmetry task with $f = 1$ filler; (b) Symmetry task with $f = 10$ filler; (c) Part-of-speech tagging. Error bars indicate $\pm 1$ SEM.

Materials. Figure 6c compares performance for the RNN and SDRNN as a function of training set size. For small data sets, the SDRNN shows an impressive advantage.

## 5 RELATED WORK IN DEEP LEARNING

Our SDRNN has an intriguing relationship to other recent developments in the deep learning literature. The architecture most closely related to SDRNN is the *fast associative memory model* of Ba et al. (2016). Like SDRNN, their model is a sequence-processing recurrent net whose hidden state is modulated by a distinct multistage subnetwork. This subnetwork has *associative weights* which are trained by a different objective than the primary task objective. Ba et al.'s focus is on using the associative weights for rapid learning over the time span of a single sequence. At the start of every sequence, the weights are reset. As sequence elements are presented and hidden state is updated, the associative weights are trained via Hebbian learning. The goal is to evoke a memory of states that arose earlier in the sequence in order to integrate those states with the current state. Although their goal is quite different than our denoising goal, Hebbian learning does enforce weight symmetry so their model might be viewed as having a variety of attractor dynamics. However, their dynamics operate in the hidden activation space, in contrast to our more computationally powerful architecture which operates in a hidden-hidden state space (i.e., the hidden representation of the RNN is mapped to another latent representation).

Graves (2016) has proposed a model for sequence processing which performs multiple steps of internal updating for every external update. Like Graves's model, SDRNN will vary in the amount of computation time needed for internal updates based on how long it takes for the attractor net to settle into a well formed state.

Turning to research with a cognitive science focus, Andreas et al. (2017) have proposed a model that efficiently learns new concepts and control policies by operating in a linguistically constrained representational space. The space is obtained by pretraining on a language task, and this pretraining imposes structure on subsequent learning. One can view the attractor dynamics of the SDRNN as imposing similar structure, although the bias comes not from a separate task or data set, but from representations already learned for the primary task. Related to language, the *consciousness prior* of Bengio (2017) suggests a potential role of operating in a reduced or simplified representational space. Bengio conjectures that the high dimensional state space of the brain is unwieldy, and a restricted representation that selects some information at the expense of other may facilitate rapid learning and efficient inference. For related ideas, also see Hinton (1990).

Finally, in a similar methodological spirit, Trinh et al. (2018) have proposed an auxillary loss function to support the training of RNNs. Although their loss is quite different in nature than our denoising loss—it is based on predicting the input sequence itself—we share the goal of improving performance on sequence processing by incorporating additional, task-orthogonal criteria.

## 6 CONCLUSIONS

Noise robustness is a highly desirable property in neural networks. When a neural network performs well, it naturally exhibits a sort of noise suppression: activation in a layer is relatively invariant to noise injected at lower layers (Arora et al., 2018). We developed a recurrent net architecture with an explicit objective of attaining noise robustness, regardless of whether the noise is in the inputs, output labels, or the model parameters. The notion of using attractor dynamics to ensure representations are well-formed in the context of the task at hand is quite general, and we believe that attractor networks should prove useful in deep feedforward nets as well as in recurrent nets. In fact, many of the deepest vision nets might well be reconceptualized as approximating attractor dynamics. For example, we are currently investigating the use of convolutional attractor networks for image transformations such as denoising, superresolution, and coloring (Thygarangan et al., 2018). Successfully transforming images requires satisfying many simultaneous constraints which are elegantly and compactly expressed in a Lyapunov (energy) function and can be approximated by deep feedforward nets. A denoising approach may also be particularly effective for transfer learning. In transfer learning, representations from one domain are applied to another, quite analogous to our SDRNN which leverages representations from one training epoch to support the next.

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

# A  MODEL INITIALIZATION

We initialize the attractor net with all attractor weights being drawn from a normal distribution with mean zero and standard deviation .01. Additionally, we add 1.0 to the on-diagonal weights of $\boldsymbol{W}^{\text{IN}}$ and $\boldsymbol{W}^{\text{OUT}}$, i.e., $W_{ii}^{\text{IN}} = 1 + \epsilon$ and $W_{ii}^{\text{OUT}} = 1 + \epsilon$  for $i \leq \min(m, n)$, where $\epsilon \sim \mathcal{N}(0, .01)$ and $m$ is the number of units in the input to the attractor net and $n$ is the number of internal (hidden) units in the attractor net.

# B  EXPERIMENTAL DETAILS

In all experiments, we chose a fixed initial learning rate with the ADAM optimizer. We used the same learning rate for $\mathcal{L}_{\text{denoise}}$ and $\mathcal{L}_{\text{task}}$. For all tasks, $\mathcal{L}_{\text{task}}$ is mean squared error. For the synthetic simulations (parity, majority, Reber, and symmetry), we trained for a fixed upper bound on the number of epochs but stopped training if the training set performance reached asymptote. (There was no noise in any of these data sets, and thus performance below 100% is indicative that the network had not fully learned the task. Continuing to train after 100% accuracy had been attained on the training set tended to lead to overfitting, so we stopped training at that point. In all simulations, for testing we use the weights that achieve the highest accuracy on the training set (not the lowest loss).

## B.1  PARITY

We use mean squared error for $\mathcal{L}_{task}$ in Parity, Majority, Reber, and Symmetry. In Parity, the learning rate is .008 for $\mathcal{L}_{\text{task}}$ and $\mathcal{L}_{\text{denoise}}$. Because the data set is noise free, training continues until classification accuracy on the training set is 100% or until 5000 epochs are reached. Training is in complete batches to avoid the noise of mini-batch training. Hidden units are tanh, and the data set has a balance on expectation of positive and negative examples. Our entropy calculation is based on discretizing each hidden unit to 8 equal-sized intervals in $[-1, +1]$ and casting the 10-dimensional hidden state into one of $10^8$ bins. For this and only this simulation, the attractor weights were trained only on the attractor loss. In all following simulations, the attractor weights are trained on both losses (as described in the main text).

## B.2  MAJORITY

The networks are trained for 2500 epochs or until perfect classification accuracy is achieved on the training set. The attractor net is run for 5 steps. Ten hidden units are used and $\sigma = 0.25$. On expectation there is a balance between the number of positive and negative examples in both the training and test sets.

## B.3  REBER

The networks are trained for 2500 or until perfect classification accuracy is achieved on the training set. We filtered out strings of length greater than 20, and we left padded strings with shorter lengths with the begin symbol, B. The training and test sets had an equal number of positive and negative examples. The architecture included $m = 20$ hidden units and $n = 40$ attractor units, and the attractor net is run for 5 steps. Without exploring alternatives, we decided to postpone the introduction of $\mathcal{L}_{\text{denoise}}$ until 100 epochs had passed.

## B.4  SYMMETRY

The networks are trained for 2500 or until perfect classification accuracy is achieved on the training set. A learning rate of .002 was used for both losses for $f = 10$ and .003 for both losses for $f = 1$. The attractor net was run for 5 iterations.

## B.5  PART OF SPEECH TAGGING

The data set had 472 unique POS tags. We preserved only the 147 most popular tags, and lumped all the others into a catch-all category, for a total of 148 tags. The catch-all category contained only

0.21% of all words. We also kept only the 20,000 most frequent words in the vocabulary, placing the rest into a catch-all category with POS also tagged as belonging to the catch-all category. An independent attractor network is used for each RNN direction. We use $m = 50$ GRU hidden units, $n = 100$ attractor units, $\sigma = 0.5$, and 15 attractor iterations. We apply a dropout of $p = 0.2$ to the output of RNN layer before projecting it into the final POS class probability layer of size $o = 148$. RNN models need to tag every word in the sequence, so averaged categorical cross-entropy is used as an optimization objective, where the target distribution is a one-hot encoding of POS tags. Attractor nets were run for 15 iterations with $\sigma = 0.5$.

