# OpenReview forum: "State-Denoised Recurrent Neural Networks"
_ICLR.cc/2019/Conference_

### Official Review · AnonReviewer1 · 2018-10-25
**Review for State-Denoised Recurrent Neural Networks**

**Rating:** 5
**Confidence:** 4

**Review:**

In this paper the authors develop the clever idea to use attractor networks, inspired by Hopfield nets, to “denoise” a recurrent neural network.  The idea is that for every normal step of an RNN, one induces an additional "dimension" of recurrency in order to create attractor dynamics around that particular hidden state. The authors introduce their idea and run some basic experiments. This paper is well written and the idea is novel (to me) and worthy of exploration.  Unfortunately, the experiments are seriously lacking in my opinion, as I believe *the major focus* of those experiments should be comparisons to other denoising / regularization techniques.

MAJOR

The point is taken that RNNs are susceptible to noise due to iterated application of the function. In my experience, countering noise (in the sense of gaussian noise added) isn’t a huge problem in practice because there are many regularization methodologies to handle it. This leads me to the point that I think the experiments need to compare across a number of regularization techniques.  The paper is motivated by discussion of noise, “noise robustness is a highly desirable property in neural networks”, and the experiments show improved performance on smaller datasets, all of which speak to regularization. So I believe comparisons with regularization techniques are pretty important here.

MODERATE

There is some motivation at the beginning of this piece, in particular about language, and does not contain citations, but should.

“Training is in complete batches to avoid the noise of mini-batch training.”  Please explain, I guess this is not a type of noise that the method handles?

What about problems that require graded responses, which is likely anything requiring integration? For example,  what happens in the majority task if the inputs were switched to a non-discrete version, where one must hold analog numbers?


MINOR

Any discussion about the (presumably dramatic) increase in training time due to the attractor dynamics unrolling + additional batching due to noise vectors (if I understood correctly)?

What are your confidence intervals over?  Presumably, we’d like to get confidence over multiple network instantiations.

Pg 1. Articulated neural network?


QUESTIONS

Does using a the ‘c’ variable as a bias instead of an initial condition really matter?

How does supervised training via eqn (4) relate to the classic training of Hopfield nets? I assume not at all, but it would be useful to clarify?

What RNN architecture did you use in the Figure 5 simulations (tanh vanilla RNN or GRU?)

---

> ### Author Response · Authors · 2018-11-27
> **Response to AnonReviewer1**
>
> Your major point is appreciated, but we worry we have misled readers by using the term 'noise' in a fast and loose manner. Certainly corruption to the hidden state due to untrained or poorly trained weights is _not_ anything close to Gaussian. We see that we have been misleading in suggesting that our denoising training procedure is designed to eliminate noise, especially in the context of training it on inputs with added Gaussian noise. What the training procedure actually does  is to establish (nonGaussian) attractor manifolds that cause a set of hidden states to be clustered together. (See our Response to AnonReviewer3 for additional details.)  It is this clustering of states where we believe the attractor dynamics are valuable.  Denoising facilitates this clustering. The advantage of clustering with attractor dynamics over something like K-Means is that (1) attractor dynamics are flexible in terms of the number and shape of clusters, (2) we can compute gradients through attractor dynamics.  While we certainly can and should conduct simulations with other regularizers, we are highly confident that they will not have the same property as our attractor net denoising.
>
> You asked whether it matters whether the 'c' variable is used as a bias rather than initial condition. It is absolutely essential for c to be a bias as it has a persistent effect on a final state. In Figure 3, treating c as a bias achieves a type of skip connection (see blue lines) that facilitates back propagation.  Finally, note that even in Hopfield nets (and certainly in Boltzmann machines), the external input must serve as a persistent bias that helps to shape energy landscapes.  We did some experiments in which c is _also_ used as the initial state, but doing so did not affect the results.
>
> The Hopfield net training algorithm (Hebbian learning) is simpler than our loss-minimizing training procedure. But the Hopfield algorithm does not support hidden attractor state.  We will note this in subsequent revisions.

---

### Official Review · AnonReviewer3 · 2018-11-04
**Interesting submission, though more analysis could help**

**Rating:** 5
**Confidence:** 3

**Review:**

I think overall I appreciate the idea behind the work. I think the work is quite novel, and it also connects to bodies of literature (hopfield networks -- attractors based and more mainstream GRU/LSTM nets). Here are some notes that I have:

1) There is a citation to anonymous 1994 -- not sure if it helps with anything. Is this work published? Why not, 1994 is quite a bit ago! I can’t see any reason why from 1994 until now this should stay anonymous.

2) Intuitively I like the idea of denoising. Though not sure exact what is denoised and towards what? In particular for hopfield networks (and I think most of the body of work that this paper points to), the idea is you have a set of sequences that you want to *memorize*. So you build a point attractor for each of this sequence, such that when starting the dynamical system in the vicinity of the point attractor (in its basin of attraction) then you converge to it (remembering the wanted sequence). Going back to this work, what is this sequence of patterns that we want to remember? More explicitly, for SDRNN you do backprop to get the h you would want and that make that a target (second loss) of the attractor net. But I'm confused about timescale. If h is not stable for a longer time, do you really converge on the attractor net ? Do we have evidence of that? Is this even meaningful early on in training, it feels like it should hurt.

3) Connecting to this, I would really love to see more analysis, going beyond measuring entropy. How do we now this is not just more capacity and the auxiliary loss just helps the optimization. Particularly since the problems are synthetic, not large scale more analysis should be possible.  How does this compare to training the simple RNN but with gaussian noise on h (to learn to be robust to it). Can we control for capacity between RNN and SDRNN?

4) To that point there is this work (not citet as far as I can tell) : https://arxiv.org/abs/1312.6026. It does have a structure somewhat similar, though none of the denoising perspective or the auxiliary loss used in this work. However the work points out that if you make the network deep in a similar way to how it was done here even though technically it is a more powerful model, gradients do not propagate well. The solution was skip connection. In the baseline that was run you do not have skip connections, and the auxiliary loss might play the role of what skip connections or a more powerful optimizer would have played.

---

> ### Author Response · Authors · 2018-11-27
> **Responses to AnonReviewer3**
>
> We anonymized the 1994 citation because one of the current paper authors was a co-author on the 1994 paper.
>
> Thank you for your spot-on summary of what denoising is intended to do by reference to Hopfield nets. Hopfield nets can take a corrupted or partial input and reconstruct the stored memory. An important property of Hopfield nets is that if two stored memories are very close, they can combine into a manifold that contains both memories (and other similar states), thereby performing a type of implicit clustering.  It is this clustering of nearby states that we leverage when we train the attractor net on the set of hidden states reached by the RNN. By mapping similar hidden states to the same attractor or attractor manifold, we impose a bias on the network to ignore small variations in the hidden state. This bias is valuable for symbolic tasks and imposes a form of regularization  early in training.
>
> You asked about time scale:  The time scale of change to the hidden state over the elements of a sequence is completely orthogonal to the time scale of attractor dynamics. In Figure 3, the time scale of hidden state evolution is represented by the columns and the time scale of the attractor net evolution is represented by the green rows of neurons.
>
> Concerning the Pascanu et al. (arXiv 1312.6026) paper: Our SDRNN and RNN+A architectures  certainly fall into the class of models with deep hidden-to-hidden transitions, as described by Pascanu. As Pascanu noted, these models do not train well without skip connections. Indeed, our approach also leverages skip connections of a sort (as represented by the blue edges in Figure 3). However, the skip connections and architecture depth are not in and of themselves sufficient:  our RNN+A fails, although it has both, whereas our SDRNN success, because it has the auxillary denoising loss. We will add citations to Pascanu.

---

### Official Review · AnonReviewer2 · 2018-11-08
**Interesting use of denoising based on attractor dynamics in RNNs, but weak experimental validation.**

**Rating:** 6
**Confidence:** 4

**Review:**

The authors propose to embed in a recurrent neural network (RNN) a multistage subnetwork that is trained to denoise its own state. This is done with an additional denoising cost term that essentially encourages the recurrent subnetwork to suppress noise during as the recurrence is unfolded in time.
The authors first demonstrate the denoising properties of this architecture, and then demonstrate its performance on a series of tasks combining it with regular tanh and GRU recurrent units.

The paper is clear and the main idea is rather interesting, but the presented experimental validations are arguably weak. The demonstration of the denoising properties of the network is rather superficial, in the sense that it does not give much insight into the functioning of the architecture, despite that presumably being the main goal of the section. In particular, it is not clear where the non-monotonic change in denoising as a function of network size comes from. Based on the attractor neural network literature that the authors cite at the beginning of the paper, it could be due to either the presence of spurious attractors, the absence of fixed-point attractors or the fact that the attractor network is trained above capacity. But the authors never go into a detailed analysis that could reveal the detailed functioning of their architecture and merely mention the hypothesis that for large networks denoising performance would decrease because of overfitting.
As for the experiments that are presented in the rest of the paper, while relevant, the types of tasks and datasets on which the proposed architecture is being tested are rather small.

Here are some more specific comments and questions:
- It would help to clarify the training procedure to explicitly mention in step 3 that training proceeds on sequences with added noise.
- It is not clear how many times in each experiments step 3 is being repeated for each mini-batch, i.e. what computational overhead is required for the training of the SDRNN compared to a regular RNN.
- It is not clear whether the recurrent neural net called RNN with attractors (RNN+A) is indeed an attractor neural network. Does the state of the network indeed always converge to an attractor?

---

> ### Author Response · Authors · 2018-11-27
> **Responses to AnonReviewer2**
>
> It would indeed be interesting to have more insight into the factors contributing to the simulations in Section 2.1 (Figure 2). However, we limited our investigation of this stand-alone attractor net because it is based on randomly placed target states in the input/output space, and this assumption is most certainly violated when the target states are constrained by task context (as they are when the attractor net is incorporated into the sequence-processing net).
>
> We have to agree with you that our experiments are modest in size, and we've done more data set exploration since the submission deadline. We have found that tasks with a nontrivial symbolic component (e.g., language processing) benefit the most, and other large-scale problems typically do not.
>
> Concerning  clarification of step 3 of the training procedure: In step 3, we give a pointer to section 2.1 which describes the denoising training procedure in detail.
>
> Concerning computational cost of denoising: As you suspect, this method adds significant more overhead to training. Our initial goal was to understand if and under what circumstances the architecture yields better generalization.
>
> Concerning RNN+A: the attractor component still has weight constraints that ensure attractor dynamics. Our main goal with RNN+A was to show that the architecture alone is inadequate to obtain good results; rather, the mix of training objectives is critical.

---

### Meta-Review · Area_Chair1 · 2018-12-18
**Promising novel idea for RNN training, with too limited experiments**

**Confidence:** 4
**Recommendation:** Reject

**Metareview:**

The paper is well written and develops a novel and original architecture and technique for RNNs to learn attractors for their hidden states (based on an auxiliary denoising training of an attractor network). All reviewers and AC found the idea very interesting and a promising direction of research for RNNs. However all also agreed that the experimental validation was currently too limited, in type and size of task and data, as in scope. Reviewers demand experimental comparisons with other (simpler) denoising / regularization techniques; more in depth experimental validation and analysis of the state-denoising behaviour; as well as experiments on larger datasets and more ambitious tasks.